# Mammalian cell display with automated oligo design and library assembly allows for rapid residue level conformational epitope mapping
Niklas Berndt Thalén[1,4], Maximilian Karlander [1,4], Magnus Lundqvist[1], Helena Persson[2], Camilla Hofström[2], S. Pauliina Turunen [2], Magdalena Godzwon[3], Anna-Luisa Volk [1], Magdalena Malm [1], Mats Ohlin [3] & Johan Rockberg [1] ✉

Precise epitope determination of therapeutic antibodies is of great value as it allows for further comprehension of mechanism of action, therapeutic responsiveness prediction, avoidance of unwanted cross reactivity, and vaccine design. The golden standard for discontinuous epitope determination is the laborious X-ray crystallography method. Here, we present a combinatorial method for rapid mapping of discontinuous epitopes by mammalian antigen display, eliminating the need for protein expression and purification. The method is facilitated by automated workflows and tailored software for antigen analysis and oligonucleotide design. These oligos are used in automated mutagenesis to generate an antigen receptor library displayed on mammalian cells for direct binding analysis by flow cytometry. Through automated analysis of 33930 primers an optimized single condition cloning reaction was defined allowing for mutation of all surface-exposed residues of the receptor binding domain of SARS-CoV-2. All variants were functionally expressed, and two reference binders validated the method. Furthermore, epitopes of three novel therapeutic antibodies were successfully determined followed by evaluation of binding also towards SARS-CoV-2 Omicron BA.2. We find the method to be highly relevant for rapid construction of antigen libraries and determination of antibody epitopes, especially for the development of therapeutic interventions against novel pathogens.

Antibodies play a critical role in many aspects of life science[1]. Molecular understanding of antibody-antigen interactions is of increasing importance as antibody-based therapies, diagnostics, and vaccines make their ways into clinical applications[2]. This knowledge of the precise epitope of an antibody can help to understand the mechanism of action and be an important tool for both antibody therapy and vaccine development processes[3–5]. Furthermore, it can be used to stratify responders from non-responders for antibody-based therapies, enabling precision medicine[6].

The golden standard of epitope mapping is by structural analysis of the antibody-antigen complex through X-ray crystallography, nuclear magnetic resonance spectroscopy (NMR) or more lately cryogenic electron microscopy (cryo-EM)[7,8]. Although these methods present a detailed structure, the contribution to the binding site of each amino acid cannot be determined since only the relative distance may be extracted from such experiments. Furthermore, the process to generate sufficient amounts of both the binding protein and target protein is a time-consuming, and sometimes uncertain process[9]. The quality and purity of each protein need to be high and scanning for optimal crystallization conditions required for X-ray crystallography as well as data collection optimization for sample determination in cryo-EM limits the use of these techniques from being swiftly used in routines[10]. As alternatives to generate a structure of the binding region, binding contribution of linear epitopes can be

¹Department Protein science, KTH—Royal Institute of Technology, Stockholm SE-106 91, Sweden. ²Science for Life Laboratory, Drug Discovery and Development Platform & School of Biotechnology, KTH-Royal Institute of Technology, Stockholm, Sweden. ³Department of Immunotechnology, Lund University, Lund, Sweden. ⁴These authors contributed equally: Niklas Berndt Thalén, Maximilian Karlander. ✉e-mail: johanr@biotech.kth.se

**Fig. 1 | A high-throughput experimental workflow enables quick and detailed scanning mutagenesis for epitope determination.** 1. Surface selection. Surface exposed amino acids are automatically extracted from antigen structure for an interaction focused library design. 2. Primer design. An exhaustive oligo optimization is undertaken to define optimal sequences for a one-reaction library construction. Resulting mutational oligos are formatted for a one click order via commercial primer synthesis company of choice. 3. SAMURAI cloning. Automated overnight cloning using designed oligos generates single mutational clones in 96-well format. 4. Affinity assessment. The constructed library is expressed on individual CHO cell surfaces and antibody binding is analyzed in a 96 well plate flow cytometer. 5. Epitope determination. Flow cytometry output is automatically processed to generate an amino acid resolution epitope presented as graphical output images. The protein structure shown in the figure was edited and rendered in PyMOL (v3.0.0) (PDB: 6YLA)[43].

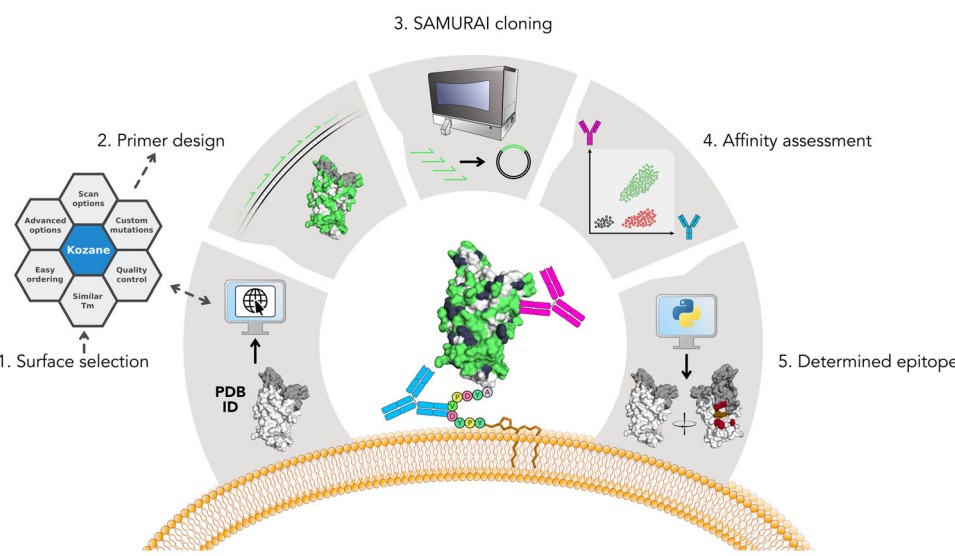

experimentally measured by fragment-based methods. These peptide-based methods are typically based on overlapping synthetic peptides (typically 9-15mer)[11] or truncated recombinantly produced larger portions of the antigen[12]. Additionally, large scale interactomes can be determined this way exploring reactivity towards both known and unknown antigens through ultra-dense peptide arrays[13] or presented as peptide libraries on *Escherichia coli* (*E.coli*)[14] or Gram-positive bacteria[15]. However, for the identification of conformational epitopes, the full tertiary construct needs to be present, which is rarely the case for peptide-based approaches.

The use of cell surface display of full-length protein mutagenesis libraries avoids the need for individual peptides to be chemically or recombinantly produced and purified. If used together with a flow cytometer, epitope residues can quickly be identified in a high throughput and comprehensive way similar to planar or bead array-based methods[12] while at the same time maintaining a relevant presentation of the epitope avoiding need for coupling of proteins to inorganic surfaces with potential damaging effects on the protein structure. Several display methods have been shown to be well suited for display of alanine mutagenesis libraries[7,16–19] and, both yeast and bacterial cell surface display have been used for epitope determination[18,20,21]. Recently, modernized methods such as yeast mating enabled high-throughput screening of antibodies targeting SARS-CoV-2[22,23] and yeast display have been used to map escape mutants on RBD[24,25]. However, expression in non-mammalian cells may limit the ability to present the correct structure of complex proteins due to limitations in folding, post translational modification (PTM) and secretion machinery in these hosts. Instead, for correct processing and display, some proteins benefit from the expression in a mammalian host[26].

For mammalian cell surface display, integral membrane proteins are often used for presenting the protein of interest on the cell surface[27–30]. However, using a transmembrane domain for protein anchoring can be a challenge since intramembrane folding is a complex endeavor for the cell[31]. As an alternative display strategy, glycosylphosphatidylinositol- (GPI) anchoring of proteins have gained attention for its benefits on protein expression and cell membrane engineering[32–35]. As opposed to integral membrane proteins, the small GPI anchor is added during PTM and is therefore not dependent on a correct protein assembly of a full transmembrane region[33]. Furthermore, GPI-anchoring leads to less membrane disruption and a higher degree of structural freedom of the expressed protein, leading to an increased solubility and function of the displayed protein[35]. Additionally, GPI anchoring enables surface expression of soluble proteins such as cytokines, increasing the throughput of alanine substitution studies.

In this study, an automated, high-throughput epitope mapping protocol was developed based on a mammalian cell surface display method that anchors the antigen on the surface of Chinese Hamster Ovary (CHO) cells via post-translational addition of GPI. With this method, the epitopes of antibodies targeting SARS-CoV-2, the causative agent of coronavirus disease 2019 (COVID-19)[36] that shortly after discovery turned into a global pandemic, were determined. SARS-CoV-2 is a coronavirus with significant homology to its predecessor, SARS-CoV-1[37]. As such, its characteristic spike glycoprotein is involved in the infection of human cells through binding to the cell receptor angiotensin-converting enzyme 2 (ACE2)[38]. This binding is mediated by a region located on the first subunit (S1) of the spike protein called the receptor binding domain (RBD). Due to this key characteristic the spike protein has been the center of diagnostic, vaccine, and therapy development[37,39]. Here, the epitopes on RBD of four antibodies targeting SARS-CoV-2 as well as the binding surface of RBD recognized by the human receptor (ACE2) were defined. For all identified epitopes, both residues broadcasting a moderate loss of binding as well as key residues that had a complete loss of binding were identified. Additionally, several automated cloning and epitope identification processes were developed to enable rapid epitope identification. This involved both high-throughput assembly of an RBD alanine mutation library in two days and software tools aiding in oligonucleotide design, addressing the shortcomings of other tools while being applicable to multiple mutagenesis methods, and downstream epitope determination. All five epitopes, identified by the presented workflow, were confirmed by either crystal structure references or competition binding assays.

## Results
### A high-throughput workflow enables fast residue level epitope determination
One of the major aims of this study was to develop a workflow for rapid and accessible epitope mapping with residue level resolution. The generated workflow is illustrated in Fig. 1 and consists of the following steps: First, an automated primer design tool (Kozane) was developed to facilitate fast and optimized design of mutagenic primers for surface exposed residues of a protein. Kozane identified surface exposed residues from a PDB file based on a user configurable threshold (Step 1 in Fig. 1). For each selected residue, Kozane designed optimized mutagenesis primers with a minimized Tm difference (Step 2 in Fig. 1) that were subsequently used in an automated SAMURAI[40] mutagenesis reaction (step 3 in Fig. 1) where each alanine mutant was individually generated. The mutagenesis step was performed in a single step 96 well format, after which individual clones for each alanine variant were acquired. Each plasmid clone was transfected into ExpiCHO™ cells and expressed on the cell surface through GPI-anchoring. Antibody binding and protein expression was determined by flow cytometry (step 4 in Fig. 1) avoiding the need for time consuming purification and quantification

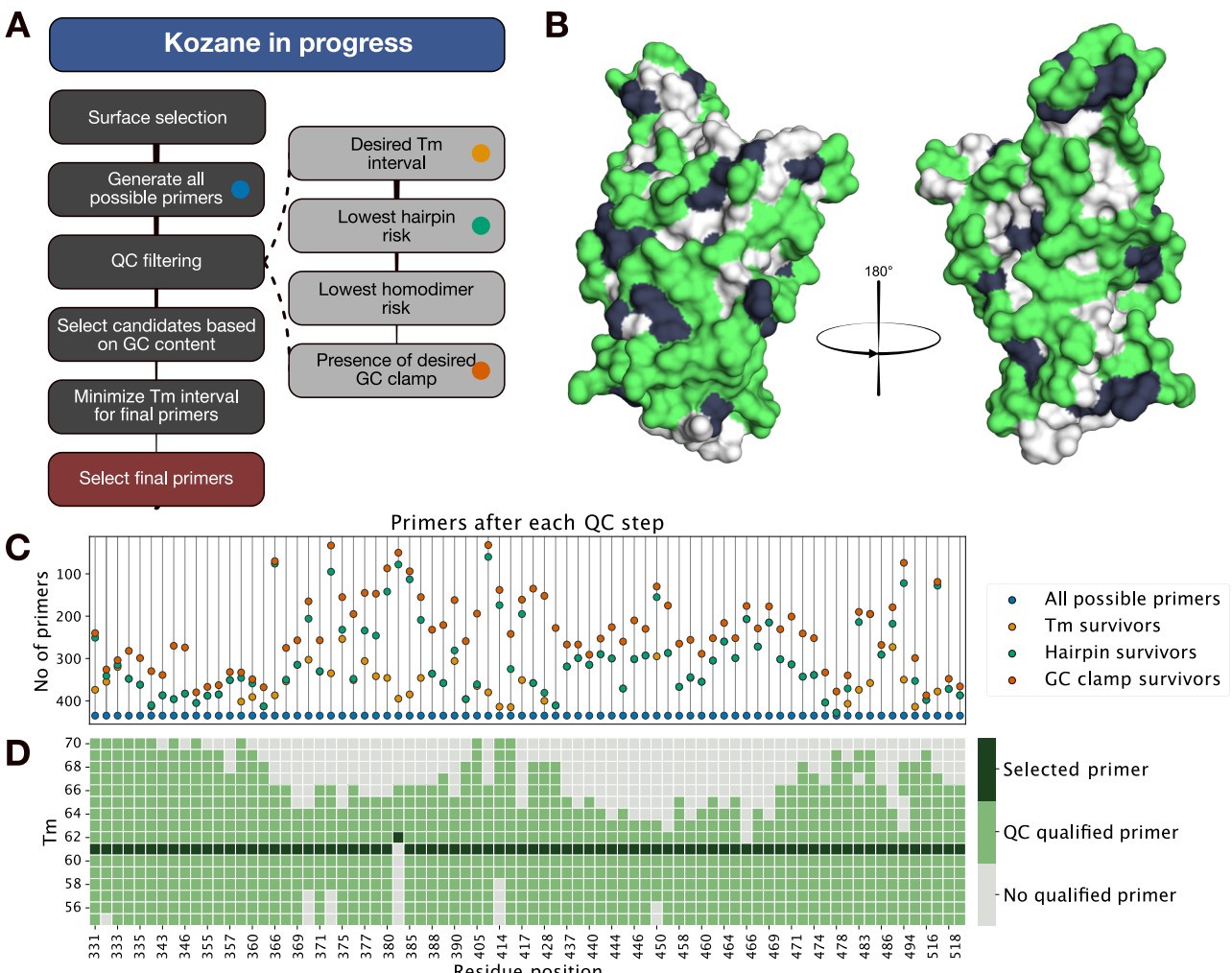

**Fig. 2 | Easy and fast optimization of primers for mutational library cloning.**
Automated design of mutagenic primers using the tailored software Kozane.
**A** Workflow. The program selects surface exposed residues and generates all possible primers based on antigen sequence to mutate user selected residues. All primers subsequently go through a series of quality control steps based on melting temperature (Tm), hairpin, homodimer formation and GC clamp. Finally, quality-verified primers, with a minimal difference in Tm, are selected based on their GC content. **B** Surface selection of SARS-CoV-2 RBD residues. Automated analysis of a SARS-CoV-2 RBD structure generated all positions with >20% RSA. After excluding alanine, glycine, cysteine, and proline (black), 78 positions (green) remained as the selected alanine mutation library. **C** Individual filtering of primers. 33930 primers (blue) representing 435 possible primers for each of the 78 mutations were generated and filtered based on TM (yellow), hairpin (green), homodimer and GC clamp (red) as illustrated. The homodimer category was not included in the figure, as no primer failed this step. **D** Final primers are selected with minimal Tm difference. Kozane selected final primers with a Tm of 61 °C- except for amino acid position 383, where the closest primer had a Tm of 62 °C. All selected primers provided the desired mutations in a one condition mutagenesis protocol.

of each individual alanine mutant. Finally, a second software tool was developed to automatically analyze the flow cytometry data of all samples (step 5 in Fig. 1) for epitope determination. This epitope determination software consisted of two steps: (1) Calculation of mean binding per expression for each individual clone and antibody combination, and (2) normalization and plotting for each tested antibody. A bar plot of the normalized mean binding per expression for every residue was generated for each individual antibody, along with a heatmap of the same values for all antibodies. Furthermore, a PyMOL script was used to automatically generate 3D images to visualize each antibody epitope. The described workflow enabled mutagenesis of 78 residues and epitope determination of four antibodies and a natural ligand, as described in more detail below.

### Automated primer design and mutagenesis with Kozane and SAMURAI

The Kozane software tool was developed to automate and optimize mutagenic oligo design for large scale cloning applications (Fig. 2A). Kozane offers the option to identify and select surface exposed residues for scanning mutagenesis of only exposed residues or manual choice of residues. All possible primers for each desired mutation were designed and filtered based on user configurable Tm interval, risk of hairpin and homodimer formation, and presence of GC clamp. All primers passing the user defined QC step for each mutation were evaluated together for minimal Tm difference for all the selected mutations. Here, PDB entry 6VW1 was used as input to identify exposed residues on RBD where Kozane identified 78 residues on RBD with >20% RSA (Fig. 2B) for mutation to alanine. This selection of surface accessible residues reduced the mutational library by 61% while retaining 73% of the protein surface area. For these 78 residues 33930 mutational primers for alanine substitution were automatically designed and filtered (Fig. 2C) resulting in a final mutagenic primer with the same Tm (61 °C) for all but one residue (62 °C) (Fig. 2D). The narrow temperature span of the mutagenic primers allowed for a one condition mutagenesis of all variants with the same temperature settings. The 78 mutagenic primers were used for alanine substitution in a 96-well format based on the SAMURAI cloning method for generating a specific user-defined site-directed mutagenic library[40]. The 78 alanine-mutated double stranded DNA inserts were then digested and ligated

**Fig. 3 | Epitope determination for CR3022 and ACE2.** Epitope comparison between cryo-ER and alanine-scanning mutagenesis by mammalian cell surface display. The color for each residue corresponds to the normalized log mean binding per expression. Gray color represents the same binding as the negative control, red and blue represent decreased and increased binding, respectively. The brown residue R355 represents a structural site of RBD where a loss of binding occurred for all antibodies tested. **A** Binding effect of all mutated residues for CR3022 and ACE2. The key-residues were K378 and Y380 for CR3022 and Y473 for ACE2. **B, C** Visual representation of the CR3022 epitope. The green area represents residues of RBD in proximity of CR3022 as determined with cryo-ER. The residues F377, K378, Y380, and H519, located within the green area, were identified as the four most contributing residues for antibody binding. The three most prominent residues K378, Y380, and F377 made up a hot spot region where K378 and Y380 almost completely remove antibody binding when mutated to alanine. **D, E** Visual representation of the ACE2 epitope. The gray area are residues of RBD in proximity of ACE2 as determined using X-ray crystallography. Residues Y449, Y473, F486, and R466 were the top four contributing residues for ACE2 binding, with Y473 identified as the most prominent residue where mutation to alanine almost completely abolished ACE2 binding towards RBD.

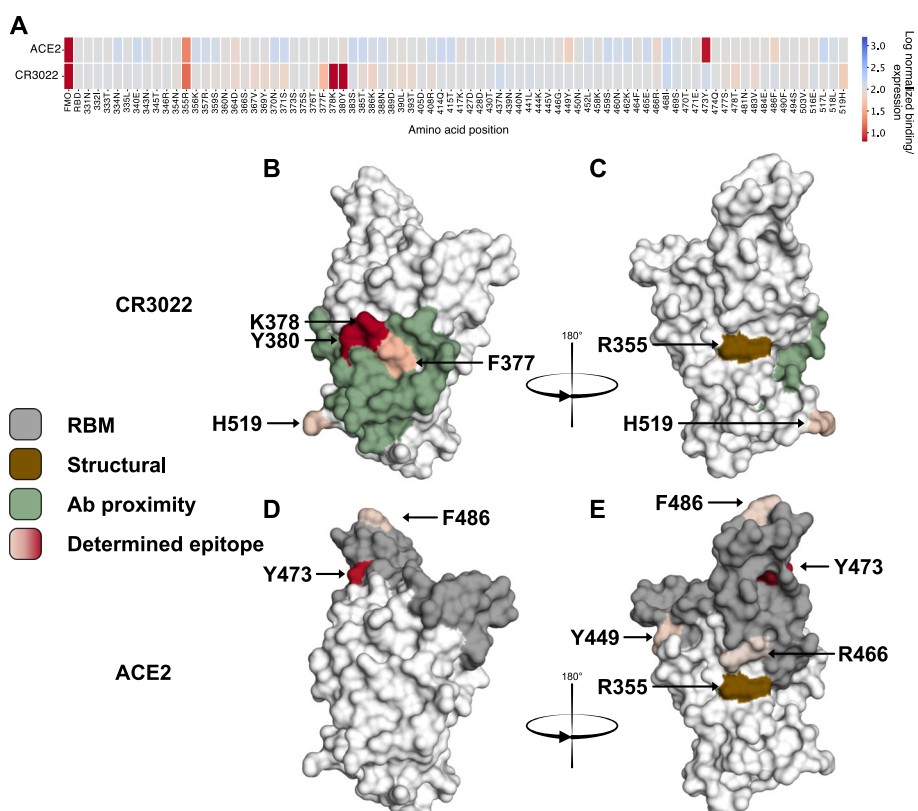

## Cell surface display epitope mapping of two reference binders for method validation

Our alanine scanning mutagenesis method was initially validated by epitope mapping of two reference binders towards SARS-CoV-2. A binding range was established by calculating binding per expression for an expressing, non-binding sample (FMO, normalized to 0), and WT RBD (normalized to 100) (Supplementary Fig. S1). This range was used to normalize each tested mutant's capacity to bind the tested binders. For the anti-SARS-CoV-2 antibody CR3022, the identified binding residues aligned within proximity of the described reference binding surface[43] (Fig. 3A–C). Mutagenesis of residues K378 and Y380 to alanine each reduced antibody binding to levels equivalent to the negative control (Fig. 3A). This aligns with the reference study where K378 was suggested as one of the main contributors for antigen binding[43]. Additionally, alanine mutants of two residues within close proximity to the key residues K378 and Y380 on RBD, F377 and H519, also decreased binding of CR3022 (Fig. 3A–C). All four of the top antibody binding residues, determined by mammalian surface display of the alanine scanning RBD library, harbor within the defined antibody binding area. In the case of the second reference binder, the SARS-CoV-2 entry receptor on human cells, ACE2, one key binding–residue, Y473, was identified by mammalian surface display (Fig. 3A, D, E). Additionally, residues F486, Y449, and R466, are all within the described reference binding surface[44], also disrupted binding when mutated to alanine (Fig. 3A, D, E).

## Epitope identification of three novel antibodies against SARS-CoV-2

Isolation of antibodies towards SARS-CoV-2 RBD and S1 was performed by affinity selection using phage display in four rounds with increasing

selection pressure applied for each round. Individual binding clones (MO176-156, MO176-301, MO176-317) were identified and re-cloned into a eukaryotic expression vector and expressed as human IgG. The affinity selected antibodies plus the reference antibody (CR3022) were evaluated in an SPR setting for determination of affinity and kinetic constants (Supplementary Fig. S2). The following dissociation constant ($K_D$) values were determined; 1.6 nM for MO176-301, 26 nM for MO176-156, 1.8 nM for MO176-317, and 24 nM for CR3022.

Surface display epitope determination of MO176-156, MO176-301, and MO176-317 against the SARS-CoV-2 RBD alanine scanning library, identified one unique and one shared epitope region between antibodies (Fig. 4). MO176-156 and MO176-301 shared the same epitope region with R466 identified as a key binding residue for both antibodies (Fig. 4A–E). Additionally, R357 and E516 acted as key binding residues for MO176-156, while N360 also contributed to binding but to a lesser degree (Fig. 4A–C). For MO176-301, R357, F464, and Y473 were identified as contributing to binding to a lesser degree than R466. These three amino acids are all within close proximity to the key binding residue R466 and compose an epitope region that is closer to the receptor binding motif (RBM) of RBD than MO176-156 (Fig. 4A, D, E). For MO176-317, an epitope distinctly different from MO176-156 and MO176-301 was identified within the RBM (Fig. 4A, F, G). Here, no residue completely depleted antibody binding, but all top identified residues highlighted an epitope in the vicinity of the displayed protruding top part of RBD (Fig. 4A, F, G) with F486 as the major contributor to antibody binding.

## Epitope binning reaffirms cell display determined epitopes but with a lower resolution

Two orthogonal methods, competition SPR and HTRF, based on epitope binning were utilized for comparison to the mammalian display epitope mapping method described here. Binding of soluble RBD was tested against immobilized ACE2 in SPR (Fig. 5A). Pre-incubation of RBD with MO176-317 decreased the binding of ACE2 to RBD whereas pre-incubation with

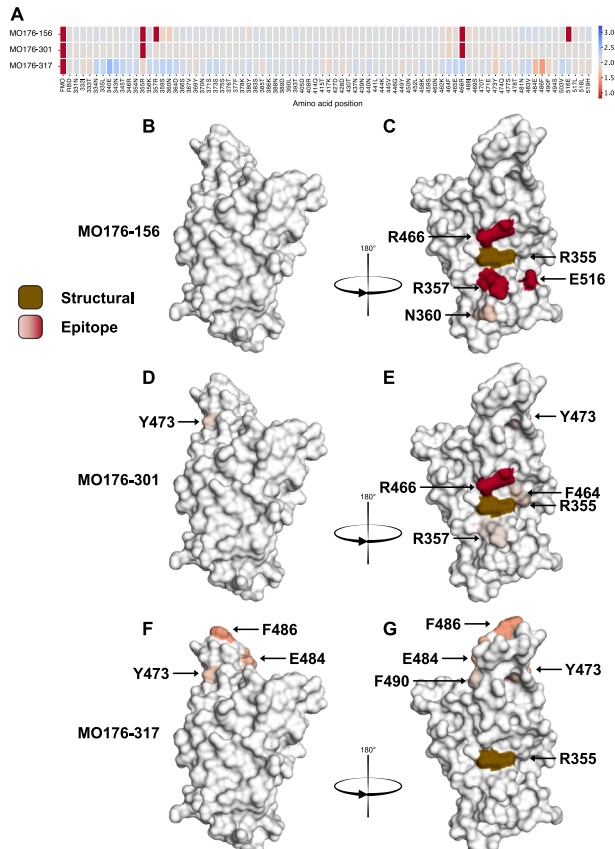

**Fig. 4 | The epitope determination of MO176-156, MO176-301, and MO176-317 showed two different epitope regions on SARS-CoV-2.** Alanine scanning mutagenesis revealed epitopes for all tested antibodies with one unique epitope that also overlaps with ACE2 (MO176-317) and two overlapping epitopes (MO176-156 and MO176-301). The color for each residue corresponds to the normalized log mean binding per expression. Gray represents the same binding as the negative control, red or blue represent decreased or increased binding, respectively. Brown residue R355 represents a structural site of RBD where a loss of binding occurred for all antibodies tested. **A** Effect of residue mutation on binding for MO176-156, MO176-301, and MO176-317. The key epitope residues, with largest decrease in binding, were R357, R466 and E516 for MO176-156 and R466 for MO176-301. The strongest binding disrupting residue for MO176-317 when mutated to alanine was F486. **B, C** Visual representation of the MO176-156 epitope. Key-binding residues R357, R466, E516 form a well defined epitope region at the center of RBD. The fourth top residue that disrupts antibody binding, N360, harbors just below the three key-residues, making up an epitope that overlaps with MO176-301. **D, E** Visual representation of the MO176-301 epitope. R466 was identified as a key-residue for antibody binding with an almost complete loss of binding when mutated to alanine. Furthermore, neighboring residues F464, R357, and Y473 were the next three in line of the four residues that most effect antibody binding. For MO176-301, residues overlap with both the ACE2 epitope (Y473) and with the MO176-156 epitope (R466, R357) were identified. **F, G** Visual representation of the MO176-317 epitope. Y473, E484, F486, and F490 were identified as the four most contributing residues for antibody binding. Here, no residue showed complete loss of binding when mutated to alanine but F486, which was also identified as important for ACE2 binding, was the most prominent one.

MO176-156 and MO176-301 had no effect (Fig. 5A). HTRF (Fig. 5B) confirmed one of the SPR results with overlapping epitopes for ACE and MO176-317 (Fig. 5C). However, in both the HTRF and surface display experiments, MO176-301 and ACE2 showed binding competition (Figs. 4A, 5C) that was not registered by the SPR epitope binning (Fig. 5A). Furthermore, HTRF showed overlapping epitopes for MO176-156 and MO176-301 (Fig. 5C), which were also identified by cell surface display (Fig. 4A). Additionally, the CR3022 epitope showed a partial overlap with

MO176-156 in the HTRF experiment but not with any of the other tested antibodies nor ACE2 (Fig. 5C).

## Binding evaluation towards the Omicron BA.2 variant of RBD confirm surface display identified epitopes

The SARS-CoV-2 variant Omicron BA.2 have 16 mutations within RBD compared to the first emerged Wuhan SARS-CoV-2 strain[45] (Fig. 6A–B). Out of the 16 Omicron BA.2 mutations, only residue 484 (mutated from glutamic acid to alanine) was identified as part of an epitope of any of the tested binders (MO176-317). MO176-156, MO176-301, CR3022 and ACE2 binding evaluation towards surface displayed Omicron BA.2 revealed no clear difference in binding compared to the RBD control, whereas MO176-317 had a strong reduction in antibody binding (Fig. 6C). This correlated well with the cell surface display epitope mapping results as E484, mutated in Omicron BA.2, was identified as one of the top contributors for MO176-317 binding (Fig. 4A, F, G).

## Discussion

Accurate and fast methods for determination of antibody epitopes are in great need[46]. Here we present a method circumventing several of the time-consuming steps (such as protein expression and purification of each protein mutant) for targeted mutagenesis and use it for rapid mammalian surface display of SARS-CoV-2 antigens (Fig. 1). The developed workflow enabled us to construct a mutational library of all surface-exposed residues and determine the corresponding epitope of five analytes within two weeks. First, elimination of buried residues in the target protein enabled a less laborious epitope identification by only selecting amino acids for mutagenesis that had more than 20% relative solubility. This drastically reduced the mutated library size from 156 clones (number of RBD residues excluding alanine, cysteine, proline, and glycine) down to 78, while still including residues with a high probability of contributing to the binding of the epitope. For even higher epitope resolution, this procedure could be complemented by a second epitope mapping round where all neighboring residues, excluded in the first round of epitope determination, are mutated to alanine. The presented surface exposure calculations rely on a pre-existing 3D structure of the target protein for the identification of exposed amino acids. The workflow is also fully compatible with recently developed protein structure prediction methods such as AlphaFold[47], which has the capacity to predict protein structures based on the amino acid sequence only. Finally, the method could be applied without any protein structure to generate a global alanine mutation library.

Although selection of only surface exposed residues greatly reduced the alanine mutation library, it can be a laborious task to generate all the 78 mutants needed for epitope mapping of RBD. Therefore, the presented primer design tool Kozane was developed and combined with an automated cloning protocol. Kozane allows for automated primer design and rapid analysis of all possible primers for specific mutations and optimizes these to fit a single reaction, according to adjustable criterias. This tool provided fast and thorough examination of all possible primers for RBD alanine scanning, enabling successful automated cloning of all constructs in one 96-well plate. Four of the mutational constructs did not yield any correct clones in the initial screening, which was hypothesized to be caused by a pipetting error when manually adding the samples for transformation. One additional cloning and transformation with no alterations was performed and the remaining mutated constructs were successfully constructed, giving a 100% successful cloning rate with the designed primers. The whole cloning process only took two days since all steps, including SAMURAI mutagenesis, solid phase cloning, ligation, and transformation, could be performed with the same settings in one plate.

Comparing the presented oligo design tool with online tools for specific methods such as Agililent QuikChange (Agilent QuikChange Primer Design) and New England Biolabs Q5 Site-Directed Mutagenesis (NEBaseChanger) clearly shows the benefits of utilizing well designed primers for a one condition cloning procedure. For conventional cloning methods, the avaliable online-tools are only giving designed primers for one mutation at a

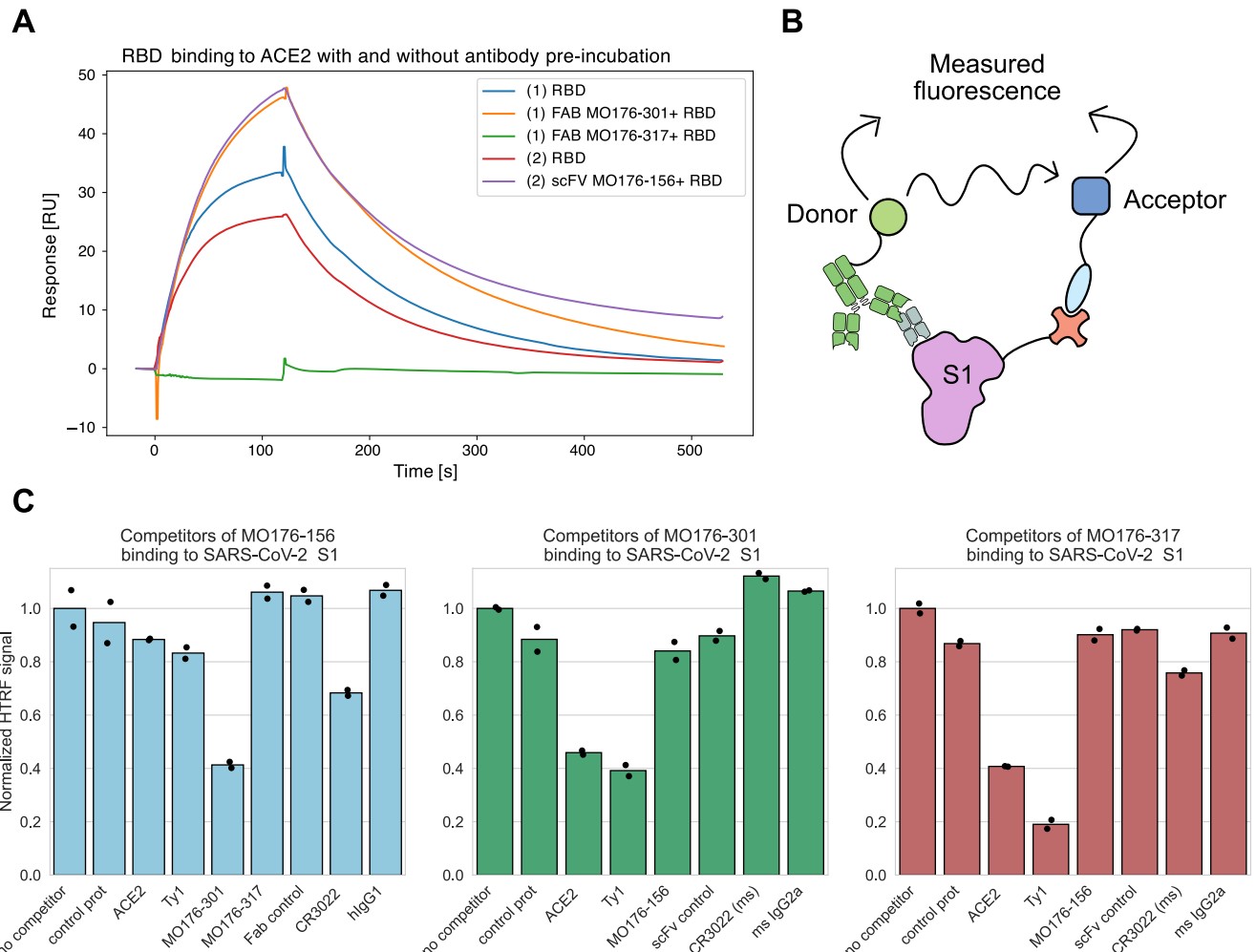

**Fig. 5 | Affinity and characteristics of human SARS-CoV-2 neutralizing antibodies. A** SPR competition assay identified competing epitopes of RBD for MO176-317 and ACE2. Pre-incubation of RBD with each of the three antibodies revealed that only MO176-317 could prevent the binding of ACE2 to RBD in solution. **B** Schematics of competitive HTRF for coarse epitope mapping towards SARS-CoV-2 S1. When in close proximity, donor molecules can excite acceptor molecules and emitted fluorescence can be measured. Measuring both donor background emission and acceptor emission enables ratiometric R values, achieving a more robust assay. If the epitope region is blocked by a competitor antibody, less binding of the primary antibody occurs and less fluorescence will be produced, thereby revealing the binding profiles of antibodies towards the target. **C** Delta R values for competitor HTRF assay for MO176-156, MO176-301, and MO176-317. In blue, antibody binding assay using HTRF of MO176-156 showed that when preincubated with MO176-301, a significant loss of binding occurred, indicating a shared epitope of the two antibodies. However, when the opposite incubation was made (in green), no epitope competition was identified for the two antibodies. Furthermore, Ty1 and ACE2 pre-incubation impacted MO176-301 binding capacity, revealing overlapping epitopes. In Red, the HTRF mapping of MO176-317 indicate that this clone shares an epitope with both ACE2 and the Ty1 nanobody.

time with different anealing temperatures since their optimal use case is for a small set of mutations rather than libraries. Additionally, each individual mutation would take 2 hours according to the manufacturer's protocols. Trials for a more high-throughput assay with the Q5® site directed mutagenesis kit have not generated sufficient number of correct clones to make it feasible for a single condition library construction[48].

Instead, the mutagenic workflow presented here, enables fast epitope determination of both future pandemic-causing pathogens, where speed and accuracy are of essence, and of clinically important targets, where detailed information of structural epitopes is of great value.

Precise epitope determination can be a difficult task since some residues might only slightly contribute to protein binding. For crystal structure epitope determination, the target residues within close proximity to the binding antibody define the epitope, resulting in a large binding area where only a fraction of the amino acids actually contributes to binding[49,50]. For the surface display epitope mapping workflow described here, the top four amino acid residues that have the largest loss of binding are automatically visualized in a 3D image of the target protein. Alternating the number of

residues to highlight in the protein structure is an easily accessible feature of the analysis script and alternating the amount of top binding contributors to show in the structure enables an easy way to review the flow cytometry results. Displaying the four top contributors is a good starting point when analyzing the epitopes of multiple binders simultaneously, as highlighted in the results within this study (Figs. 3, 4). For a more thorough epitope determination it is advisable to carefully review each residue that show a loss of binding compared to control in the flowcytometry, making sure that they are within a surface reachable by the binding protein and not structurally disrupting residues. As highlighted here, both key residues and residues that contribute to a lesser degree are easily identified through alanine mutation scanning through mammalian surface display. Additionally, although GPI-anchoring has several benefits shown in this study it might not act as the most suitable display methods when studying epitope parts that are within a multi-pass membrane since the native membrane sites might play a role in correct protein assembly. However, we do believe that for any extracellular domain that is not part of a domain going through the membrane, GPI-anchoring of the extracellular part is an advisable choice for surface display.

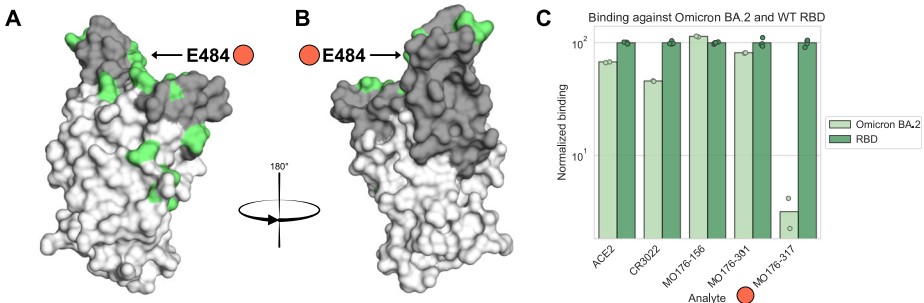

**Fig. 6 | Cross-reactivity to Omicron BA.2.** The anti-RBD antibodies and ACE2 protein were tested for binding against RBD of the Omicron BA.2 variant. **A**, **B** Visual representation of Omicron BA.2 mutations. Omicron BA.2 mutations are in green. E484, marked with red, was the only residue previously identified as part of an epitope mutated in Omicron BA.2. **C** Binding capacity to Omicron BA.2

compared to wildtype. All RBD binders except MO156-317 had very similar binding to both RBD variants whereas MO176-317 had drastically impaired binding to Omicron BA.2. This antibody was the only one with an identified epitope residue overlapping with a mutated residue in Omicron BA.2 (E484, red in **A** and **B**), which in Omicron BA.2 happens to be mutated to alanine.

Two reference RBD-interacting proteins, CR3022[43] (PDB:6YLA) and ACE2[44] (PDB:2AJF), with available crystal structures acted as proof-of-concept in this study. For both reference mappings, all residues that showed a clear decrease in binding when mutated to alanine were located within the binding area (Fig. 3), verifying the presented cell surface display method. Furthermore, the identified residues highlight the few actual contributors to the epitope within a surface that is close to the binding protein, exhibiting the need to not only determine a binding region based on distance, but actual binding contribution of each residue.

After experimental validation of our epitope mapping model, the epitopes of three novel antibodies binding RBD were determined in the same manner as for the two references. Interestingly, all three of the determined antibody epitopes had resides that overlapped with the determined ACE2 epitope (Fig. 3A, D, and Fig. 4). Most prominently are the key residues Y473 and F486 that are shared between ACE2 and MO176-317. This information is also revealed in the HTRF experiments were MO176-317, MO176-301, and MO176-156 show competition with ACE2, although in a less detailed manner. However, for the SPR competition assays only competition between ACE2 and MO176-317 was identified. This strong competition between MO176-317 and ACE2 is clear when reviewing the surface display epitope mapping since the key residues Y473 and F486 for ACE2 is also shared as the binding region for MO176-317. Furthermore, an additional shared epitope was identified for MO176-301 and MO176-156 (Fig. 4) just below the RBM of RBD. This shared epitope was also identified using HTRF, however, interestingly only when MO176-301 is added first in the competition assay (Fig. 5C). This could be explained by the difference in affinity and size between the two constructs since the Fab MO176-301 had around 15 times higher affinity compared to the scFv MO176-156 (Supplementary Fig. S2), as well as the larger size of a Fab compared to a scFv. These differences may influence the potential for simultaneous binding depending on the order of binding, highlighting a limitation in epitope determination by the HTRF assay. Further limitations of a less detailed epitope mapping method can be seen on the HTRF competition results between CR3022 and MO176-156. Here, a slight competition is seen between the two, but it can be hard to interpret the reason or the level of overlapping epitopes. With the surface display epitope mapping, E519 is clearly a crucial residue for MO176-156, and this residue is in close proximity to the stretch between H519, Y380, and K378 where the CR3022 antibody binds and hence some steric hindrance between MO176-156 and CR3022 could occur. This detailed amino acid level epitope identification broadens the understanding of these two antibody binding sites. Besides the detailed information gathered for all epitopes through the presented method, all epitopes identified are conformational, showing the importance to look at whole protein domains when determining epitopes.

One important feature when identifying tertiary structure displayed epitopes through alanine mutations, are the potential structural residues that are not part of the binding site. Here, the epitope mapping of cell surface

displayed RBD showed low levels of both binding and cell surface expression for the alanine mutation of amino acid R355. When full protein domains are processed through a mammalian secretory machinery it undergoes several control mechanisms for correct folding and processing. Due to the low levels of surface expression of the R355A mutated RBD (Supplementary Fig. S1, Supplementary Data 2) and the low binding to different antibodies, we postulate that it is an amino acid of importance for correct structure of the RBD domain. Furthermore, R355 has been reported to play an important role in stabilizing the spike protein through a salt-bridge formation between R355 and D398[51]. Identification of structurally important residues as R355 adds to the understanding of SARS-CoV-2 and is only identified when scanning mutagenesis is performed at full protein or protein domains such as in the presented method. Furthermore, it highlights the benefit of choosing a suitable expression host for a protein, as the same results acquired from for example bacterial expression could have been blamed on the host. However, for complete verification of structural residues, two binding proteins targeting different epitopes are needed.

Additionally, all four antibodies and ACE2 were tested for binding capacity against the Omicron BA.2 variant of SARS-CoV-2. The results showed that MO176-317 is not a suitable antibody for Omicron BA.2 targeting. Interestingly, the low degree of binding of MO176-317 to Omicron BA.2 RBD, could be predicted based on the detailed mammalian surface display epitope mapping alone, where E484, a mutated residue in Omicron BA.2, was identified as an important site for MO176-317 interaction (Fig. 4F, G).

In conclusion, our automated method for library generation and GPI-anchored surface display of whole protein domains discovered both key-residues that completely disrupted binding and residues contributing to the epitope of two reference antibodies targeting Sars-CoV-2 that aligned with the available reference structure. Additionally, three novel antibody epitopes were mapped in detail and aligned to additional experimental epitope identification methods. This fast epitope identification was made possible by an automated surface selection and primer design tool that enabled the construction of a mutagenic library in two days. We anticipate that the presented method can make future epitope determinations more accessible and accurate.

## Methods
### Antibody generation
Phage display was performed using the SciLifeLib synthetic library of human scFvs and Fab fragments, based on the IGHV3-23 and IGKV1-39 genes and constructed using Kunkel mutagenesis[52]. Biotinylated SARS-CoV-2 S protein RBD (SPD-C82E9, Acro Biosystem (Newark, DE, USA) or SARS-CoV-2 S1 protein (SIN-C82E8, Acro Biosystem) were used as antigens during four rounds of selections on magnetic streptavidin-coated beads, using gradually increased selection pressure with succeeding rounds[53]. Individual clones were screened for target binding in enzyme-

linked immunosorbent assay (ELISA) and surface plasmon resonance (SPR), resulting in a shortlist of 20 sequence unique clones being converted to human IgG1. Among these, clones MO176-156, MO176-301 and MO176-317, of special interest to this study, can be found. Gene sequences encoding the heavy (VH) and light (VL) chain variable domains of the different clones were PCR amplified and inserted into modified versions of the pIgG vector[54] (GenBank: MK988448.1), using the In-Fusion HD Plus Cloning Kit (638909, Clontech (Mountain View, C, USA). Proteins were expressed by transient transfection of human embryonic kidney cells in suspension. Transfection of plasmid DNA into ExpiHEK293™FTM cells (A14527, ThermoFisher (Carlsbad, CA, USA)) was performed using the ExpiFectamine™ 293 Transfection Kit (A14525, ThermoFisher (Carlsbad, CA, USA)). After five days of cultivation at 37 °C, 8% CO2, 70% relative humidity, and 125 rpm, the culture supernatants were retrieved, and IgG was purified by protein A magnetic beads (88846, Pierce (Rockford, IL, USA)).

## Implementation of an automated mutagenesis tool

An automated primer design tool (Kozane - www.kozane.app) was developed to facilitate fast and optimized design of DNA oligos for targeted mutagenesis. The software tool was designed with a set of features and user parameters described below. Kozane enables determination of residues to mutate based on surface exposure through the xssp web server API, using the DSSP output[55,56]. Solvent accessible surface area values from the DSSP result are subsequently recalculated into RSA values with the theoretical reference values from Tien et al. [57]. For every residue chosen for mutation, all possible primer candidates are compiled. Each candidate will contain a user definable number of bases on each side of the mutation while still being within the user defined maximum total primer length. Subsequently, all candidates pass through a series of quality filtering steps. The first quality filtering step is melting temperature (Tm), based on a user-defined accepted Tm interval. Tm is calculated using Primer3[58] and is adjusted for mismatches. Candidates passing the Tm step are evaluated for hairpins and homodimers using Primer3. If a hairpin or homodimer Tm is predicted to be above a user defined threshold, the candidate is discarded. The final step checks each candidate for a suitable GC-clamp. This is defined as either the last base or 2-3 of the last 5 3' bases being cytosines or guanines. If no candidates pass a filtering step, a handful of the most-suited candidates will be rescued, and a warning will be given on the results page.

Candidates for each residue passing these quality filtering steps are considered together to minimize the Tm interval between residues. If a single Tm has primers for each residue, this Tm will be chosen. If more than one Tm has primers for each residue, the Tm with the least extreme (furthest from 50%) GC-content for a single primer is chosen. If no Tm has primers for each residue, primers with different Tm's are chosen with a minimal difference.

## Surface exposure calculations

Protein data bank (PDB) entry 6VW1 for SARS-CoV-2 chimeric receptor-binding domain complexed with its receptor human ACE2 was used as input reference in the surface exposure online tool of Kozane with calculations performed on chain A and with a relative solvent accessibility (RSA) threshold of 0.2[55–57]. If a short sequence was missing from the PDB file and the RSA value could not be determined, it was included as surface exposed. Apart from surface exposed amino acid cysteine, proline, glycine, and alanine were excluded from mutation due to structural concerns.

## Surface display vector- and alanine substitution library construction

Primers for alanine substitution were generated in the online primer design tool Kozane. The following settings were used; Tm_low = 50, Tm_high = 70, 5' ext = /5Phos/, type = alanine scan, primer direction = R, flank = 7, maxlength = 45. The Kozane tool generated individual primers for all mutations with a Tm between 61–62 °C (supplementary data 1). Primers were obtained from Integrated DNA Technologies (Leuven, Belgium) in

5 μM concentration, dissolved in water in a 96 well plate. Forward 5'biotinylated primers and reverse primers for insert amplification were designed and obtained from Integrated DNA Technologies (Leuven, Belgium) as freeze-dried samples (supplementary data 1).

The plasmid vector for GPI anchored protein display was based on the PCDNA3.3 vector with added HA-tag and GPI region downstream of the display protein cloning site. The SARS-CoV-2 RBD (330-583, Wuhan strain[45]) was PCR amplified from the SARS-CoV-2 spike protein[59] and inserted into the GPI-display vector via restriction cloning. For Omicron BA.2 surface display construction, a geneblock of the Omicron BA.2 variant was ordered from Integrated DNA Technologies (Leuven, Belgium) and cloned into the HA-GPI surface expression vector through restriction cloning.

The SAMURAI cloning method was used for the construction of individual alanine scanning mutants[40], with an updated PCR condition to optimize for single mutation settings. The mutation PCR was performed in two steps. For the extension with mutagenic primers the following cyclic conditions were used: denaturation 98 °C 8 s, anneal 65 °C 25 s, extension 72 °C 30 s. 10 cycles were performed, with an initial denaturation temperature set to 98 °C for 30 s and final extension 72 °C for 5 min, followed by steady temperature at 4°. After this, 1 μl of 5 μM reverse extension primer were added and the following PCR conditions were used: no initial denaturation, anneal 65 °C 25 s, extension 72 °C 30 s. Thermocycling was performed for 5 rounds with no initial denaturation and a final extension of 72 °C for 5 min, followed by 4° storage. For the cloning of the generated mutated inserts, solid phase cloning was used to prepare insert for ligation in a Magnatrix 8000 (NorDiag AS, Oslo, Norway)[41].

## Epitope determination

Transfection of display vectors was performed according to manufacturer's protocol into 2.5 ml of ExpiCHO™ cells (A29133, Thermo Fisher Scientific (Carlsbad, CA, USA). 24 h post transfection RBD control cultivations were counted and based on their mean value the volume to take out for all alanine mutants was determined ($0.5 \times 10^6$ viable cells). Cells were washed twice through 300xG centrifugation for 5 min and resuspended in 200 μl PBSB (1x phosphate buffer saline + 1% bovine serum albumin fraction V (810531, Sigma-Aldrich (Saint-Louis, MO, USA)). Next, 75 μl primary antibodies were added (ACE2: 0,5 ng/μl, CR3022: 0,01 ng/μl, MO176-156, MO176-301, MO176-317: 0,1 ng/μl, in PBSB). Rabbit anti-HA antibody (HA88-342, Invitrogen (Carlsbad, CA, USA)) was used at a 1:10 000 dilution. After 1 h with 60° turning on a rotor table in RT, cells were spun down and cleaned in the same manner as before. Subsequently, secondary antibodies were added, Alexa anti-rabbit 488 and Alexa anti-human 647 (A32731 and A21445, Invitrogen (Carlsbad, CA, USA)) with a 1:1000 dilution in PBSB. Cells were then kept in the dark on ice for 1 h. After the same wash step as described earlier, cells were resuspended in 200 μl PBSB and run in a Cytoflex flow cytometer (C09748, Beckman Coulter (Indianapolis, IN, USA)). For assessment of binding capacity, a fluorescence minus one (FMO) control for each run was used as a normalization parameter together with the unmutated RBD control. Python scripts were used to analyze the flow cytometry data. For each sample a homogenous, expressing cell population was gated out. For the remaining events the antibody binding value was divided by the surface expression value and the arithmetic mean of the quotients was calculated. After calculating mean values for each sample their values were normalized with the FMO set to 0 and the RBD set to 100. The normalized values were plotted in a heatmap. Finally, the four most prominent binding disrupting positions were highlighted in the protein structure in PyMol (PDB structure 6YLA[43]) with the hue of the color corresponding to the normalized value. The scripts were designed as general tools to be used for any epitope mapping with this method[60]. The FlowCal package[61] was used for gating and plotting. Other packages used were matplotlib[62], numpy[63], pandas[64], scipy[65], and seaborn[66]. For package versions, reproducibility, and further information see the GitHub repository for the data analysis (https://github.com/mkarla/epitope_mapping).

## Kinetic measurements

Affinity and kinetic constants of the IgG antibodies were assessed using a Biacore T200 instrument (Cytiva, Uppsala, Sweden) and single cycle kinetics (SCK) approach. An anti-human Fab antibody mix (28958325, Cytiva, Uppsala, Sweden), functioning as a capture ligand, was immobilized through EDC/NHS amine coupling chemistry onto all four surfaces of a Series S sensor chip CM5-S (BR1005-30, Cytiva, Uppsala, Sweden) according to the manufacturer's recommendations. Antibodies were injected and captured onto the chip surface. Three channels were used to capture different antibodies, whereas channel 1 was used as a reference surface. Dilution series of five concentrations of SARS-CoV-2 RBD, were prepared in HBS buffer supplemented with 0.05% Tween20 buffer and sequentially injected over the chip surfaces. More specifically, a concentration range of 0.2–20 nM was used for MO176-301 and MO176-317, whereas 1.25–20 nM was used for M0176-156 and CR3022. Following a dissociation phase, the surfaces were regenerated with 10 mM Glycine-HCl, pH 2.5. All measurements were performed at 25°C with a flowrate of 30 μl/min. Response curve sensorgrams were obtained after removing the reference channel's response and a reference cycle (running buffer instead of antigen). Reaction rate kinetics constants were calculated using the Biacore T200 (Cytiva, Uppsala, Sweden) evaluation software 3.1 using the 1:1 Langmuir binding model.

## Epitope binning using homogenous time resolved fluorescence (HTRF)

ScFv and Fab antibody fragments were expressed in *E. coli* TOP10 and purified using Protein A magnetic beads (88846, Pierce, Rockford, IL, USA)[53] or HisPur Cobalt affinity resin (89966, Pierce (Rockford, IL, USA)). The purified antibody fragments, diluted in PPI Europium detection buffer (Cisbio, #61DB9RDF (Codolet, France)) to give 10 nM final concentration, were assessed for binding to monomeric SARS-CoV-2 S1 protein (poly-HisAvi-tagged, biotinylated; S1N-C82E8, Acro Biosystems (Newark, DE, USA) at 10 nM final concentration in HTRF assay casted to 384-well assay plates (3842, Corning Costar (Kennebunk, ME, USA)). For competitive binding to S1 epitopes, recombinant human ACE2 (10108-H08H, Sino-Biological (Beijing, China)), S1-binding antibody fragments scFvs, Fabs, human IgG1 CR3022 (srbd-mab1, InVivoGen (Toulouse, France)) or mouse IgG2b CR3022 (Ab01680-3.0, Absolute Antibodies (Wilton, UK)) were preincubated with monomeric S1 in 5-fold excess concentration, or Ty1 alpaca derived nanobody[67] at 100 nM final concentration, at room temperature for one hour before adding the primary antibody to be detected. Similarly, isotype-matched human and mouse IgG, scFv and Fab fragments targeting irrelevant targets, as well as recombinant His-tagged SARS-CoV-2 main protease were used as control competitors at 50 nM final concentration. Detection of binding was enabled either through europium-conjugated anti-FLAG antibody (Cisbio, 61FG2KLA (Codolet, France)) for scFv, or anti-kappa antibody (Cisbio #61KAPKAA (Codolet, France)) for Fab, both conjugates functioning as donor molecules to an acceptor, streptavidin-conjugated XL665 (Cisbio #610SAXL (Codolet, France)). Plates were incubated at room temperature for two hours before measuring fluorescence with Envision plate reader (Perkin Elmer, Waltham, MA, USA) at 615 nm (acceptor background fluorescence) and 665 nm (binding signal). The 665 nm value was divided with the 615 nm value to retrieve a ratiometric R-value and subtracted with reagent background to count delta-R for each sample. All samples were assayed in duplicates. Assays were repeated twice and mean ± SD of HTRF delta-R signal relative to non-competed condition were plotted.

## Epitope competition assay using SPR

A MASS-16 instrument (1862615, Bruker (Hamburg, Germany)), a system that quantifies molecular interactions in real time using surface plasmon resonance-based detection, was used to analyze the ability of scFv to interfere with the binding of SARS-CoV-2 RBD to ACE2. A High Capacity Amine Sensor chip (1862614, Bruker (Hamburg, Germany)) was immobilized with streptavidin (434302, ThermoFisher Scientific (Waltham, MA, USA)) (50 μg/ml diluted in 10 mM sodium acetate buffer pH 5.0; flow rate: 10 μl/min; time of immobilization: 6 min) to a level of approximately 1000 RU followed by binding of 50 nM biotinylated ACE2 (10108-H08H-B, SinoBiological (Beijing, China)) (flow rate: 10 μl/s; time of binding: 2 min) to the immobilized streptavidin. The reference spot on the chip only carried streptavidin. The binding assay was performed in Dulbecco's PBS (AM9625, Invitrogen (Waltham, Massachusetts, USA) containing 0.01% Tween 20 as follows: 60 nM RBD was pre-incubated with a scFv or a fab (200-300 nM), or just with buffer for 40 min at room temperature. Thereafter the mixtures were injected over the sensor chip for 2 min, followed by a 6 min dissociation phase (flow rate of 30 μl/s). The sensor chip was regenerated by treatment with 1 M magnesium chloride solution (#M1028, Sigma Aldrich (St Louis, MO, USA)).

## Statistics and reproducibility

For the epitope determination each alanine substitution variant was tested with each binder once. This was chosen due to the large number of samples individually tested and for the large number of data points collected for each sample. For evaluation of the different binders capacity to bind Omicron BA.2 the sample size for WT RBD binding was four and the sample size for Omicron BA.2 binding was two. The HTRF epitope binning experiments were carried out with technical duplicates.

## Reporting summary

Further information on research design is available in the Nature Portfolio Reporting Summary linked to this article.

## Data availability

Raw flow cytometry data generated for this manuscript can be obtained from Figshare: https://doi.org/10.6084/m9.figshare.23057480 Numerical data used to plot graphs in Figs. 2–6 is available on Figshare: https://doi.org/10.6084/m9.figshare.25868659 All other data is available from the corresponding author on reasonable request.

## Code availability

Software for primer design and surface exposure calculation (Kozane) is available as online tool: https://www.kozane.app The source code is available from the corresponding author on reasonable request. Python scripts for analysis of flow cytometry data generated for epitope mapping using this method are available at Figshare: https://doi.org/10.6084/m9.figshare.23057159 Current versions of these scripts are available on GitHub: https://github.com/mkarla/epitope_mapping.

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

## Acknowledgements

We wish to thank Juni Andrell at SciLifeLab (Eukaryotic Protein Production (EPP) cell lab, Stockholm University) for the production of RBD used for the generation of neutralizing antibodies. Furthermore, we wish to thank Nevan Krogan and his lab for providing the SARS-CoV-2 spike protein. Additionally, we wish to thank Gerald McInerney for providing the Ty1 nanobody. The work was funded by Knut and Alice Wallenberg Foundation, ScilifeLab, Swedish Foundation for Strategic Research (SSF), Swedish innovation agency Vinnova through GeneNova, AAVNova, CellNova and AdBIOPRO.

## Author contributions

J.R., A.V., and N.B.T. initiated the study. N.B.T., M.L., and M.K. designed experiments related to cloning of the mutation library and the mammalian surface epitope mapping. N.B.T. carried out experiments related to cloning of the mutation library and the mammalian surface epitope mapping. H.P., C.H., P.T., M.G., and M.O. designed, constructed and produced the antibodies used in the study, conducted competition SPR, kinetic SPR, and performed the HTRF epitope mapping. M.L. constructed the Kozane software and M.K. constructed the automated epitope calculation scripts. N.B.T. and M.K. wrote the manuscript. N.B.T., M.K., M.L., A.V., H.P., C.H., P.T., M.G., M.M., M.O., and J.R. reviewed the manuscript. J.R. was the main supervisor for the project.

## Funding

## Competing interests

The authors declare no competing interests.
