## [Peer review file · Communications Biology]

Reviewers' comments:

Reviewer #1 (Remarks to the Author):

The authors established and used a toolbox of methods for epitope mapping of antibodies that bind discontinuous epitopes. They developed software tools for identification of surface exposed residues that are used for alanine scanning mutagenesis thereby reducing the number of positions to be mutated significantly. They also established a mutagenesis primer creation tool and used a previously described method that is amenable to automation for cloning of the alanine scanning variants. Interestingly, the authors used mammalian surface display for presentation of model protein ACE2 and derived variants. Upon transient transfection they gated for ACE2 displaying cells using antibodies against the amended HA epitope and checked for binding of several antibodies, including two of which the epitope is already known from structural analysis. These are interesting data that merit publication.

Minor points: The part describing binding assessment and display normalization could be better explained and two example flow cytometry plots could be presented in the results section for a binding and a non-binding alanine replacement in comparison. The authors mention in the discussion that there might also exist structurally disrupting residues that when replaced by alanine could result in significant structural changes that might lead to misleading results in an epitope mapping experiment. I wonder whether additional controls could be implemented in future screens addressing this potential issue that go beyond reduced target protein display as indirect indication of reduced stability as in the case of the R355A variant.

Typos:

Results: "Tm difference(Step 2 in figure 1" introduce a space after difference

Discussion section. Replace flow cytometry by flow cytometry.

"Supplementary Data are available at NAR online" NAR online may be a wrong source.

Funding: remove the xxxxx!

Reviewer #2 (Remarks to the Author):

The use of mammalian display to carry out epitope mapping is not particularly novel, having been described in the following review:

Maes, S., et al. (2023). "Deep mutational scanning of proteins in mammalian cells." *Cell Rep Methods* 3(11): 100641, which contains a significant number of cited papers.

The following review, describing different methods to epitope map antibody binding sites should be cited:

Dang, X., et al. (2023). "Epitope mapping of monoclonal antibodies: a comprehensive comparison of different technologies." *MAbs* 15(1): 2285285.

The use of yeast mating for epitope mapping described in the following papers should be described and cited:

Younger, D., et al. (2017). "High-throughput characterization of protein-protein interactions by reprogramming yeast mating." *Proc Natl Acad Sci U S A* 114(46): 12166-12171.

Engelhart, E., et al. (2022). "Massively multiplexed affinity characterization of therapeutic antibodies against SARS-CoV-2 variants." *Antib Ther* 5(2): 130-137.

The following papers also addressed the use of yeast display to map RBD epitopes and should be cited:

Francino-Urdaniz, I. M., et al. (2021). "One-shot identification of SARS-CoV-2 S RBD escape mutants using yeast screening." *Cell Rep*: 109627.

Starr, T. N., et al. (2021). "Prospective mapping of viral mutations that escape antibodies used to treat COVID-19." *Science*.

The use of the GPI linked approach is fine for single pass membrane proteins. However, it is unlikely to work for multipass membrane proteins, such as GPCRs. These limitations should be discussed.

Reviewer #3 (Remarks to the Author):

This manuscript describes a combinatorial method for rapid mapping of discontinuous epitopes using mammalian cell display with automated oligo design and library assembly. The authors constructed the method and then identified the epitopes of two reference binders and three novel antibodies to the receptor binding domain (RBD) of SARS-CoV-2 and SARS-CoV-2 Omicron BA.2. While the authors successfully determined the epitopes on RBDs, the novelty and inventive step of this research is unclear.

1. The potential novelty of this research seems to be the development of automated primer design system Kozane for plasmid library construction. However, advantages or novelty of the Kozane compared to many other primer design tools already developed were not examined in this manuscript. The authors should compare the performance of Kozane (e.g. reliability and success rate of library construction) with other tools and clearly indicate the advantages.

2. The advantages of the epitope mapping technique used in this research is unclear. The global RBD alanine scanning mutagenesis on about 190 residues have reported to analyze the epitope of antibodies (DOI: 10.1186/s13073-021-00985-w). The shotgun alanine-scanning of antibody epitope on RBD have been also reported (DOI: 10.1016/j.celrep.2021.109784). Moreover, the deep mutational scanning of ACE2-binding region on RBD have been reported (DOI: 10.1016/j.cell.2020.08.012). Compared to these previous studies, the present technique seems to lack comprehensiveness and has the limitation that structures of RBD are required.

3. As to why the mammalian cell display was used, the authors said “expression in non-mammalian cells may limit the ability to present the correct structure of complex proteins due to limitations in folding, post translational modification (PTM) and secretion machinery in these hosts.” However, the RBD is virus-derived protein without PTM and have displayed on yeast cell surface and phage (DOI: 10.3389/fimmu.2022.935573, 10.1016/j.xpro.2021.100869, 10.1016/j.chom.2020.11.007, 10.1016/j.celrep.2021.109627, 10.1016/j.cell.2020.08.012, 10.1038/s41598-023-50450-4, 10.3389/fmicb.2022.968036). In several of these papers, epitopes of antibodies have been successfully analyzed. In other words, there is little advantage to using mammalian cell display when RBD is the antigen.

Reviewer #1 (Remarks to the Author): The authors established and used a toolbox of methods for epitope mapping of antibodies that bind discontinuous epitopes. They developed software tools for identification of surface exposed residues that are used for alanine scanning mutagenesis thereby reducing the number of positions to be mutated significantly. They also established a mutagenesis primer creation tool and used a previously described method that is amenable to automation for cloning of the alanine scanning variants. Interestingly, the authors used mammalian surface display for presentation of model protein ACE2 and derived variants. Upon transient transfection they gated for ACE2 displaying cells using antibodies against the amended HA epitope and checked for binding of several antibodies, including two of which the epitope is already known from structural analysis. These are interesting data that merit publication. Minor points: The part describing binding assessment and display normalization could be better explained and two example flow cytometry plots could be presented in the results section for a binding and a non-binding alanine replacement in comparison. The authors mention in the discussion that there might also exist structurally disrupting residues that when replaced by alanine could result in significant structural changes that might lead to misleading results in an epitope mapping experiment. I wonder whether additional controls could be implemented in future screens addressing this potential issue that go beyond reduced target protein display as indirect indication of reduced stability as in the case of the R355A variant. Typos: Results: "Tm difference(Step 2 in figure 1" introduce a space after difference Discussion section. Replace flow cytometry by flow cytometry. "Supplementary Data are available at NAR online" NAR online may be a wrong source. Funding: remove the xxxxx!	Author response: We wish to thank reviewer 1 for input that will increase the quality of our presented study. We have added an introduction sentence in the result section regarding binding assessment with an added figure for further clarification (line 389). Additionally, we have extended our discussion surrounding structural residues and thoughts on controls to verify misleading results (line 592). All the typos have also been fixed.
Reviewer #2 (Remarks to the Author):	Author response (1): All of the references mentioned are now added to the text with

(1) The use of mammalian display to carry out epitope mapping is not particularly novel, having been described in the following review: Maes, S., et al. (2023). "Deep mutational scanning of proteins in mammalian cells." Cell Rep Methods 3(11): 100641, which contains a significant number of cited papers. The following review, describing different methods to epitope map antibody binding sites should be cited: Dang, X., et al. (2023). "Epitope mapping of monoclonal antibodies: a comprehensive comparison of different technologies." MABs 15(1): 2285285. The use of yeast mating for epitope mapping described in the following papers should be described and cited: Younger, D., et al. (2017). "High-throughput characterization of protein-protein interactions by reprogramming yeast mating." Proc Natl Acad Sci U S A 114(46): 12166-12171. Engelhart, E., et al. (2022). "Massively multiplexed affinity characterization of therapeutic antibodies against SARS-CoV-2 variants." Antib Ther 5(2): 130-137. The following papers also addressed the use of yeast display to map RBD epitopes and should be cited: Francino-Urdaniz, I. M., et al. (2021). "One-shot identification of SARS-CoV-2 S RBD escape mutants using yeast screening." Cell Rep: 109627. Starr, T. N., et al. (2021). "Prospective mapping of viral mutations that escape antibodies used to treat COVID-19." Science. (2) The use of the GPI linked approach is fine for single pass membrane proteins. However, it is unlikely to work for multipass membrane proteins, such as GPCRs. These limitations should be discussed.	additional sections added both in the introduction (lines 79, 103) and the discussion (lines 463, 478). Author response (2): The limitations of GPI-anchoring for epitope mapping of multi-pass epitopes are added within the discussion (line 535).
Reviewer #3 (Remarks to the Author): This manuscript describes a combinatorial method for rapid mapping of discontinuous epitopes using mammalian cell display with automated oligo design and library assembly. The authors constructed the method and then	Author response (1): We wish to thank reviewer 3 for the remarks surrounding unclarity of benefits within our presented study since it is of uttermost importance to us that the advantageous steps such as speed and usability should be clear and available to the reader. To the best of our knowledge no

identified the epitopes of two reference binders and three novel antibodies to the receptor binding domain (RBD) of SARS-CoV-2 and SARS-CoV-2 Omicron BA.2. While the authors successfully determined the epitopes on RBDs, the novelty and inventive step of this research is unclear.

1. The potential novelty of this research seems to be the development of automated primer design system Kozane for plasmid library construction. However, advantages or novelty of the Kozane compared to many other primer design tools already developed were not examined in this manuscript. The authors should compare the performance of Kozane (e.g. reliability and success rate of library construction) with other tools and clearly indicate the advantages.

2. The advantages of the epitope mapping technique used in this research is unclear. The global RBD alanine scanning mutagenesis on about 190 residues have reported to analyze the epitope of antibodies (DOI: 10.1186/s13073-021-00985-w). The shotgun alanine-scanning of antibody epitope on RBD have been also reported (DOI: 10.1016/j.celrep.2021.109784). Moreover, the deep mutational scanning of ACE2-binding region on RBD have been reported (DOI: 10.1016/j.cell.2020.08.012). Compared to these previous studies, the present technique seems to lack comprehensiveness and has the limitation that structures of RBD are required.

3. As to why the mammalian cell display was used, the authors said “expression in non-mammalian cells may limit the ability to present the correct structure of complex proteins due to limitations in folding, post translational modification (PTM) and secretion machinery in these hosts.” However, the RBD is virus-derived protein without PTM and have displayed on yeast cell surface and phage (DOI: 10.3389/fimmu.2022.935573, 10.1016/j.xpro.2021.100869, 10.1016/j.chom.2020.11.007, 10.1016/j.celrep.2021.109627, 10.1016/j.cell.2020.08.012, 10.1038/s41598-023-50450-4, 10.3389/fmicb.2022.968036). In

publicly available tools can generate individual primers for multiple mutation points with a common Tm. We extended our discussion around available cloning methods and their available primer tools with similar functionality to Kozane, and highlighted that they do not have the same depth (line 495).

Author response (2): Highlighting the benefits of our presented epitope mapping is critical and the speed, usability, and convenient workflow needs to be clearly presented. Therefore, we have further clarified the benefits both in introduction and in the discussion part of the paper and added clarification that a crystal structure is not needed for the method but when available it adds benefits for epitope determination as for all other methods (lines 123, 463, 592).

Author response (3): We wish to thank reviewer 3 for a relevant point of discussion. As suggested, RBD can be expressed in both bacterial and yeast host. However, as described in the majority of the references RBD is a glycoprotein and the glycosylations affect the binding interaction to ACE2 (DOI: 10.3389/fimmu.2022.935573). Other proteins, such as C5, which can not be expressed properly in other hosts than mammalian, would likely have served as good examples of the benefits of mammalian display. RBD, in the end, was chosen because of its clinical relevance and the availability of newly developed antibodies in need of epitope mapping.

several of these papers, epitopes of antibodies have been successfully analyzed. In other words, there is little advantage to using mammalian cell display when RBD is the antigen.	
--	--

REVIEWERS' COMMENTS:

Reviewer #1 (Remarks to the Author):

The authors addressed my critical points in their revision of the manuscript. It can be now recommended for publication.

Reviewer #2 (Remarks to the Author):

This is suitable for publication now

Reviewer #3 (Remarks to the Author):

The author addressed all my comments. I recommend publication in the journal.